statistics

confidence interval, intrinsic credibility, prior-data conflict, *p*-value, replication, significance test

**Author for correspondence:**
Leonhard Held
e-mail: leonhard.held@uzh.ch

# The assessment of intrinsic credibility and a new argument for $p < 0.005$

## Leonhard Held

Center for Reproducible Science (CRS), Epidemiology, Biostatistics and Prevention Institute (EBPI), University of Zurich, Hirschengraben 84, 8001 Zurich, Switzerland

 LH, 0000-0002-8686-5325

The concept of intrinsic credibility has been recently introduced to check the credibility of 'out of the blue' findings without any prior support. A significant result is deemed intrinsically credible if it is in conflict with a sceptical prior derived from the very same data that would make the effect just non-significant. In this paper, I propose to use Bayesian prior-predictive tail probabilities to assess intrinsic credibility. For the standard 5% significance level, this leads to a new *p*-value threshold that is remarkably close to the recently proposed $p < 0.005$ standard. I also introduce the credibility ratio, the ratio of the upper to the lower limit (or *vice versa*) of a confidence interval for a significant effect size. I show that the credibility ratio has to be smaller than 5.8 such that a significant finding is also intrinsically credible. Finally, a *p*-value for intrinsic credibility is proposed that is a simple function of the ordinary *p*-value and has a direct frequentist interpretation in terms of the probability of replicating an effect. An application to data from the Open Science Collaboration study on the reproducibility of psychological science suggests that intrinsic credibility of the original experiment is better suited to predict the success of a replication experiment than standard significance.

## 1. Introduction

The so-called replication crisis of science has been discussed extensively within the scientific community [1,2]. One aspect of the problem is the widespread misunderstanding and misinterpretation of basic statistical concepts, such as the *p*-value [3,4]. This has led to a major rethink and new proposals for statistical inference, such as to lower the threshold for statistical significance from the traditional 0.05 level to 0.005 for claims of new discoveries [5,6]. The proposal would lead to considerably fewer, but possibly more reliable, scientific claims and has created a lot of discussion in the scientific community. The shortcut '$p < 0.005$' has even been shortlisted and highly

commended in the 2017 Statistic of the Year competition by the Royal Statistical Society, see https://bit. ly/2yRSWuZ.

Two arguments for this step are provided in Benjamin *et al.* [6]: the first is based on the Bayes factor, the second is based on the false discovery rate. Both arguments are actually not new. Edwards *et al.* [7] have already emphasized in 1963 that the evidence of *p*-values around 0.05 against a point null hypothesis, as quantified by the Bayes factor, is much smaller than one would naively expect: 'Even the utmost generosity to the alternative hypothesis cannot make the evidence in favour of it as strong as classical significance levels might suggest'. Likewise, Staquet *et al.* [8] have already argued in 1979 that the false positive rate 'could be considerably reduced by increasing the sample sizes and by restricting the allowance made for the $\alpha$ error, which should be set to a 1% level as a minimum requirement'. Benjamin *et al.* [6] therefore propose to lower the threshold for statistical significance to 0.005 and to declare results with $0.05 > p > 0.005$ as 'suggestive', emphasizing the need for replication.

In this paper, I provide a new argument for this categorization into three levels of evidence. The approach is based on the concept of intrinsic credibility [9], a specific reverse-Bayes method to assess the credibility of claims of new discoveries. I propose to base the assessment of intrinsic credibility on the Box [10] check for a prior-data conflict and show that the traditional dichotomization of *p*-values into 'significant' and 'non-significant' naturally leads to a more stringent threshold for intrinsic credibility. For the standard 5% significance level, the new *p*-value threshold is 0.0056, remarkably close to the proposed $p < 0.005$ standard. I thus provide a new argument for the 0.005 threshold for claims of new discoveries without any prior support from preceding studies.

To assess intrinsic credibility based on a confidence interval rather than a *p*-value, I propose the credibility ratio, the ratio of the upper to the lower limit (or *vice versa*) of the confidence interval for a significant effect size. I show that the credibility ratio has to be smaller than 5.8 to ensure that a significant finding is also intrinsically credible. In §2, I provide a brief summary of the Analysis of Credibility and the concept of intrinsic credibility. The latter is central to the derivation of a threshold for intrinsic credibility, as outlined in §3.

Lowering the threshold of statistical significance is only a temporary response to the replication crisis [11]. A more radical step would be to abandon significance thresholds altogether [12], leaving *p*-values as a purely quantitative measure of the evidence against a point null hypothesis. In this spirit, I extend the concept of intrinsic credibility and propose in §4 the *p*-value for intrinsic credibility, $p_{IC}$. This new measure provides a quantitative assessment of intrinsic credibility—without any need for thresholding—and has a useful interpretation in terms of the probability of replicating an effect [13]. Intrinsic credibility is thus directly linked to replication, a topic of central importance in the current debate on research reproducibility [14]. Section 5 describes an application to data from the Open Science Collaboration study on the reproducibility of psychological science [15] which suggests that intrinsic credibility of the original experiment is better suited to predict the success of a replication experiment than standard significance. I close with some discussion in §6.

# 2. Analysis of credibility

Reverse-Bayes approaches allow the extraction of the properties of the prior distribution needed to achieve a certain posterior statement for the data at hand. The idea to use Bayes's theorem in reverse originates in the work by IJ Good [16,17] and is increasingly used to assess the plausibility of scientific claims and findings [18–21]. Matthews [22,23] has proposed the Analysis of Credibility, a specific reverse-Bayes method to challenge claims of 'significance'; see Matthews [9] for more recent developments.

Analysis of Credibility is based on a conventional confidence interval of level $\gamma$, say, for an unknown effect size $\theta$ with lower limit $L$ and upper limit $U$, say. In the following, I assume that both $L$ and $U$ are symmetric around the effect estimate $\hat{\theta} \sim N(\theta, \sigma^2)$ (assumed to be normally distributed) and that both are either positive or negative, i.e. the effect is significant at significance level $\alpha = 1 - \gamma$. Matthews [22,23] proposed assessing the credibility of a statistically significant finding by computing from the data a sufficiently sceptical prior distribution for the effect size $\theta \sim N(0, \tau^2)$, that—combined with the information given in the confidence interval for $\theta$—results in a posterior distribution which is just non-significant at level $\alpha$, i.e. either the $\alpha/2$ or the $1 - \alpha/2$ posterior quantile is zero. The sufficiently sceptical prior thus describes how sceptical we would have to be to not find the apparently positive effect estimate convincing.

The required variance $\tau^2$ of the sufficiently sceptical prior is a function of the variance $\sigma^2$ (the squared standard error, assumed to be known) of the estimate $\hat{\theta}$, the corresponding test statistic $t = \hat{\theta}/\sigma$, and $z_{\alpha/2}$, the $1 - \alpha/2$ quantile of the standard normal distribution:

$$\tau^2 = \frac{\sigma^2}{t^2/z_{\alpha/2}^2 - 1}, \tag{2.1}$$

where $t^2 > z_{\alpha/2}^2$ is required for significance at level $\alpha$. Equation (2.1) shows that the prior variance $\tau^2$ can be both smaller or larger than $\sigma^2$, depending on the value of $t^2$. If $t^2$ is substantially larger than $z_{\alpha/2}^2$, then $\tau^2$ will be relatively small, i.e. a relatively tight prior is needed to make the significant result non-significant. If $t^2$ is close to $z_{\alpha/2}^2$ (i.e. the effect is 'borderline significant'), then $\tau^2$ will be relatively large, i.e. a relatively vague prior is sufficient to make the significant result non-significant.

It can also be shown [22] that the limits $\pm S$ of the equi-tailed credible interval of the sufficiently sceptical prior at level $\gamma$ are given by

$$S = \frac{(U - L)^2}{4\sqrt{UL}}, \tag{2.2}$$

where $S$ is called the *scepticism limit* and the interval $[- S, S]$ the *critical prior interval*. Note that (2.2) holds for any level $\gamma$, not just for the traditional 95% level.

Two examples of the Analysis of Credibility are shown in figure 1. Both are based on a confidence interval of width 3, but with different location ($\hat{\theta} = 2.5$ and $11/6 = 1.83$, respectively). Each figure has to be read from right to left: To obtain a 95% posterior credible interval with lower limit 0 (shown in green), the 95% confidence interval for the unknown effect size $\theta$ (shown in red) has to be combined with the sufficiently sceptical prior with variance (2.1) (shown in blue).

In this paper, I focus on claims of new discoveries without any prior support. To assess the credibility of such 'out of the blue' findings, Matthews [9] suggested the concept of intrinsic credibility, declaring an effect as intrinsically credible if it is in conflict with the sufficiently sceptical prior that would make the effect non-significant. This can be thought of as an additional check to ensure that a significant effect is not spurious. Specifically, Matthews [9] declares a result as intrinsically credible at level $\alpha$, if the effect estimate $\hat{\theta}$ is outside the critical prior interval, i.e. $|\hat{\theta}| > S$. He shows that, for confidence intervals at level $\gamma = 0.95$, this is equivalent to the ordinary two-sided $p$-value being smaller than 0.0127. I refine the definition of intrinsic credibility in the following §3 based on the Box [10] prior-predictive approach, leading to the more stringent $p$-value threshold 0.0056 for intrinsic credibility at the 5% level. The result shown in figure 1a is thus intrinsically credible ($p = 0.0011 < 0.0056$) whereas the one in figure 1b is not ($p = 0.017 > 0.0056$).

# 3. A new threshold for intrinsic credibility

Matthews' check for intrinsic credibility compares the size of $\hat{\theta}$ with the scepticism limit (2.2), so does not take the uncertainty of $\hat{\theta}$ into account. He compares the estimate $\hat{\theta}$ with the (sufficiently sceptical) prior distribution, not with the corresponding prior-predictive distribution. However, the use of tail probabilities based on the prior-predictive distribution is the established way to check the compatibility of the data and the prior [10,18]. See the discussion and the rejoinder of [10, pp. 404–430] for further aspects of the prior-predictive model criticism approach. An alternative approach to check the compatibility between prior and data has been recently proposed by de Carvalho *et al.* [24].

In the following I will apply the approach by Box [10] for the assessment of prior-data conflict based on the prior-predictive distribution, with the perhaps slightly unusual feature that the prior has been derived from the data. The maximum-likelihood prior [25], the max-compatible prior [24] and adaptive priors [26,27] are other examples of data-based prior distributions. I argue that there is nothing intrinsically inconsistent in investigating the compatibility of a prior, defined through the data, and the data itself, extending an argument by Cox [28, Section 5.10] to the reverse-Bayes setting.

The Box [10] check for prior-data conflict is based on the prior-predictive distribution, which is in our case normal with mean zero and variance $\tau^2 + \sigma^2$ [29, Section 5.8]. The procedure is based on the test statistic $t_{\text{Box}} = \hat{\theta}/\sqrt{\tau^2 + \sigma^2}$ and the (two-sided) tail probability $p_{\text{Box}} = \Pr(\chi^2(1) \geq t_{\text{Box}}^2)$ as the corresponding upper tail of a $\chi^2$-distribution with one degree of freedom. Small values of $p_{\text{Box}}$ indicate a conflict between the sufficiently sceptical prior and the data.

Now suppose we fix the confidence level at the conventional 95% level, i.e. $\gamma = 0.95$. Intrinsic credibility at the 5% level (i.e. $p_{\text{Box}} < 0.05$) can then be shown to be equivalent to the requirement

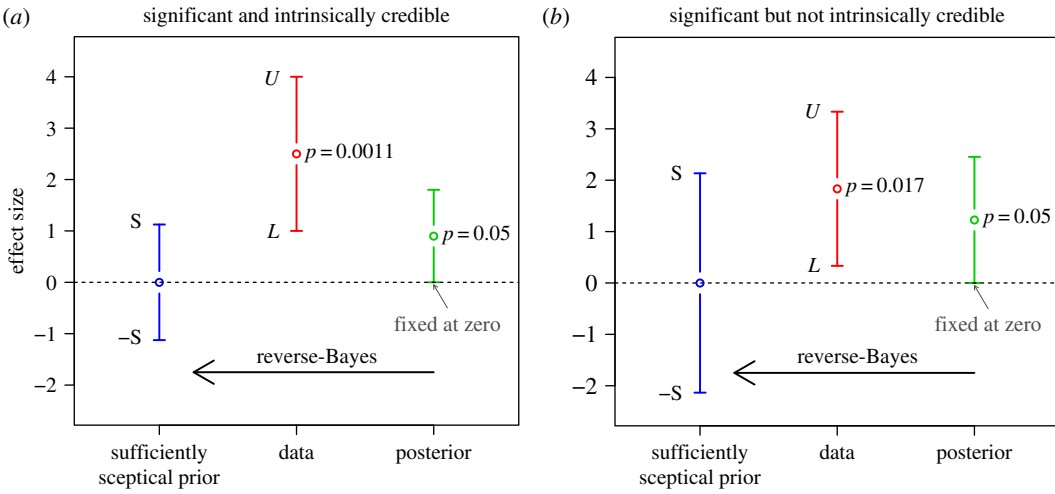

**Figure 1.** Analysis of intrinsic credibility for two confidence intervals $[L, U]$ at level $\gamma = 95\%$. In the first example, there is a conflict between the sufficiently sceptical prior and the data and the significant result is intrinsically credible at the 5% level ($L = 1$, $U = 4$, credibility ratio = 4, $p_{IC} = 0.021$). In the second example, there is less conflict between prior and data and the significant result is not intrinsically credible at the 5% level ($L = 1/3$, $U = 10/3$, credibility ratio = 10, $p_{IC} = 0.09$). The credibility ratio will be described further in §3 while the $p$-value $p_{IC}$ for intrinsic credibility will be introduced in §4.

$p < 0.0056$ for the ordinary two-sided $p$-value. To derive this result, note that with equation (2.1), we have $\tau^2 + \sigma^2 = \sigma^2/(1 - z_{\alpha/2}^2/t^2)$ and so $t_{\text{Box}}^2 = t^2 - z_{\alpha/2}^2$. The requirement $t_{\text{Box}}^2 > z_{\alpha/2}^2$ for intrinsic credibility at level $\alpha$ then translates to

$$t^2 \geq 2z_{\alpha/2}^2. \tag{3.1}$$

This criterion is to be compared with the traditional check for significance, which requires only $t^2 \geq z_{\alpha/2}^2$. It follows directly that the threshold

$$\alpha_{IC} = 2\left\{1 - \Phi\left(t = \sqrt{2}z_{\alpha/2}\right)\right\}, \tag{3.2}$$

here $\Phi(\cdot)$ denotes the cumulative standard normal distribution function, can be used to assess intrinsic credibility based on the ordinary two-sided $p$-value $p$: If $p$ is smaller than $\alpha_{IC}$, then the result is intrinsically credible at level $\alpha$. For $\alpha = 0.05$, we have $t = \sqrt{2} \cdot 1.96 = 2.77$ and the threshold (3.2) turns out to be $\alpha_{IC} = 0.0056$, as claimed above. For other significance levels, we will obtain other intrinsic credibility thresholds. For example, Clayton & Hills [30, Section 10.1] prefer to use 90% confidence intervals 'on the grounds that they give a better impression of the range of plausible values'. Then $\alpha = 0.1$ and we obtain the intrinsic credibility threshold $\alpha_{IC} = 0.02$.

Figure 2 compares the new threshold with the one obtained by Matthews [9, appendix A.4] (using $t = 1.272\, z_{\alpha/2}$) for values of $\alpha$ below 10%. The Matthews threshold for intrinsic credibility is larger than the proposed new threshold (3.2), because it compares the effect estimate $\hat{\theta}$ with the prior distribution (with variance $\tau^2$), not with the prior-predictive distribution (with variance $\tau^2 + \sigma^2$).

Intrinsic credibility can also be assessed based on the confidence interval $[L, U]$, rather than the $p$-value $p$. If both $L$ and $U$ are either positive or negative, then $t_{\text{Box}}^2$ can be written in terms of $L$ and $U$,

$$t_{\text{Box}}^2 = z_{\alpha/2}^2 \frac{4UL}{(U - L)^2}, \tag{3.3}$$

see appendix for a derivation. The requirement $t_{\text{Box}}^2 \geq z_{\alpha/2}^2$ for intrinsic credibility is then equivalent to require that the *credibility ratio* $U/L$ (or $L/U$ if both $L$ and $U$ are negative) fulfils

$$\frac{U}{L} \leq d = 3 + 2\sqrt{2} \approx 5.8. \tag{3.4}$$

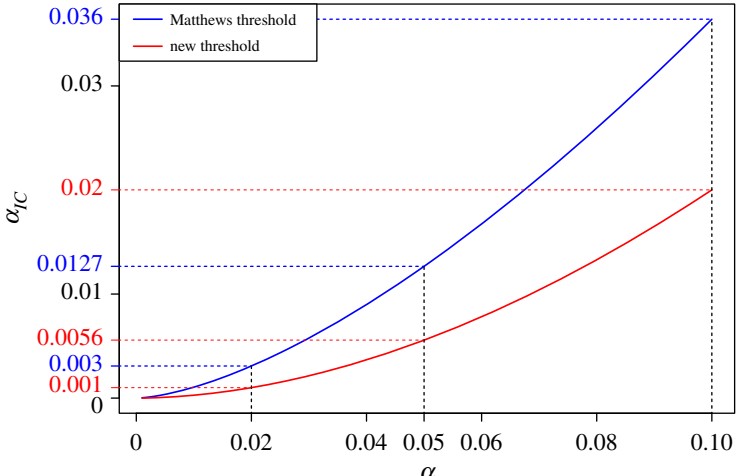

**Figure 2.** The threshold for intrinsic credibility of significant results as a function of the conventional $\alpha$ level. The blue line corresponds to the proposal by Matthews [9]. The red line is the proposed new threshold.

To derive the cut-point $d$ in equation (3.4), set $U = L\,d$. The requirement $t^2_{\text{Box}} = z^2_{\alpha/2}$ then reduces to

$$1 = \frac{4UL}{(U-L)^2} = \frac{4d}{(d-1)^2},$$

a quadratic equation in $d$ with $d = 3 + 2\sqrt{2}$ as solution.

Thus, there is a second way to assess intrinsic credibility based on the ratio of the limits of a 'significant' confidence interval *at any level* $\gamma$: if the credibility ratio is smaller than 5.8 than the result is credible at level $\alpha = 1 - \gamma$. For example, the credibility ratio is 4 in figure 1*a* and 10 in figure 1*b*, so the result shown in figure 1*a* is intrinsically credible at the 5% level, but the one in figure 1*b* is not.

If the sufficiently sceptical prior is available, then a third way to assess intrinsic credibility is to compare the prior variance $\tau^2$ to the data variance $\sigma^2$. Comparing equation (2.1) with equation (3.1), it is easy to see that intrinsic credibility is achieved if and only if the sufficiently sceptical prior variance $\tau^2$ is not larger than the variance $\sigma^2$ of the effect estimate $\hat{\theta}$. With this in mind, we see immediately that the first result shown in figure 1*a* is intrinsically credible ($\tau^2 < \sigma^2$), whereas the one shown in figure 1*b* is not ($\tau^2 > \sigma^2$).

# 4. A *p*-value for intrinsic credibility

A disadvantage of the dichotomous assessment of intrinsic credibility described in the previous section is the dependence on the confidence level $\gamma = 1 - \alpha$ of the underlying confidence interval. However, there is a way to free ourselves from this dependence. In analogy to the well-known duality of confidence intervals and standard *p*-values, I propose to derive the confidence level $\gamma^{\star} = 1 - p_{IC}$, say, that just achieves intrinsic credibility, i.e. where equality holds in (3.1). This defines the *p*-value for intrinsic credibility $p_{IC} = 1 - \gamma^{\star}$, which provides a quantitative assessment of intrinsic credibility. The *p*-value for intrinsic credibility $p_{IC}$ can also be used to assess intrinsic credibility as described in §3: if $p_{IC} \leq \alpha$, then the result is intrinsically credible at level $\alpha$.

The *p*-value $p_{IC}$ for intrinsic credibility can be derived by replacing $\alpha_{IC}$ with $p$ and $\alpha$ with $p_{IC}$ in equation (3.2) and then solving for $p_{IC}$:

$$p_{IC} = 2\left[1 - \Phi\left(\frac{t}{\sqrt{2}}\right)\right]. \tag{4.1}$$

Here $t = \Phi^{-1}(1 - p/2)$ is the standard test statistic for significance where $p$ is the ordinary two-sided *p*-value. Figure 3 shows that the *p*-value $p_{IC}$ for intrinsic credibility is considerably larger than the ordinary *p*-value $p$, particularly for small values of $p$. For example, the two confidence intervals shown in figure 1 have ordinary *p*-values $p = 0.0011$ (figure 1*a*) and $p = 0.017$ (figure 1*b*), while the corresponding *p*-values for intrinsic credibility are $p_{IC} = 0.021$ and $p_{IC} = 0.09$, respectively.

It is also possible to define a one-sided version of the *p*-value for intrinsic credibility, which turns out to be just half as large as the two-sided one, see Held [31] for a derivation in the more general context of

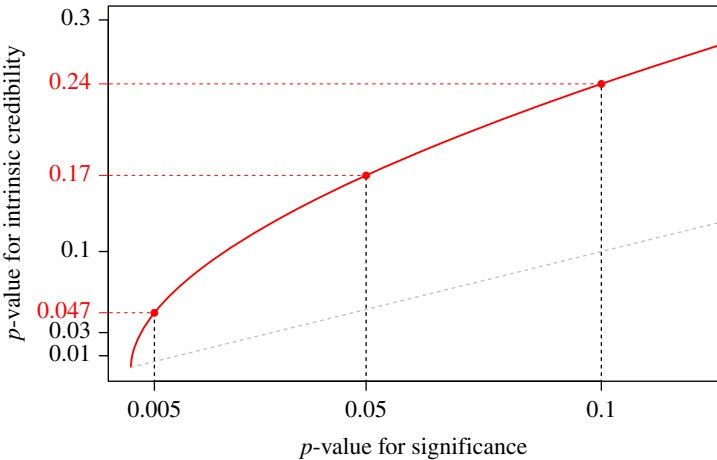

**Figure 3.** The *p*-value for intrinsic credibility as a function of the *p*-value for significance. The grey dashed line is the identity line.

replication studies. The same relationship also exists of course between one- and two-sided ordinary *p*-values, which implies that figure 3 is also valid for the corresponding one-sided *p*-values. In the following, we will only use the two-sided *p*-value for intrinsic credibility, as defined in (4.1).

There is a direct and useful interpretation of $p_{IC}$ in terms of the probability of replicating an effect, $p_{rep}$ [13], i.e. the probability that an identically designed but independent replication study will give an estimated effect $\hat{\theta}_2$ in the same direction as the estimate $\hat{\theta}_1 = \hat{\theta}$ from the current (first) study. To see this, note that under an initial uniform prior the posterior for $\theta$ is $\theta \,|\, \hat{\theta}_1 \sim N(\hat{\theta}_1, \sigma^2)$. This posterior now serves as the prior for the mean of the (unobserved) estimate $\hat{\theta}_2 \,|\, \theta \sim N(\theta, \sigma^2)$ from the second (hypothetical) study, where we assumed the two studies to be identically designed, having equal variances $\sigma^2$. This leads to the prior-predictive distribution $\hat{\theta}_2 \,|\, \hat{\theta}_1 \sim N(\hat{\theta}_1, 2\sigma^2)$ and the *p*-value for intrinsic credibility (4.1) can be seen to be twice the probability that the second study will give an estimate $\hat{\theta}_2$ in the opposite direction to the estimate $\hat{\theta}_1$ of the first study:

$$
\begin{aligned}
p_{IC} &= 2\left[1 - \Phi\left(\frac{t}{\sqrt{2}}\right)\right] \\
&= 2\Phi\left(\frac{-t}{\sqrt{2}}\right) \\
&= 2\Phi\left(\frac{0 - \hat{\theta}_1}{\sqrt{2}\sigma}\right) \\
&= 2\Pr(\hat{\theta}_2 \le 0 \,|\, \hat{\theta}_1 > 0).
\end{aligned}
$$

If $\hat{\theta}_1 < 0$, then $p_{IC} = 2\Pr(\hat{\theta}_2 \ge 0 \,|\, \hat{\theta}_1 < 0)$.

The probability $\Pr(\hat{\theta}_2 \le 0 \,|\, \hat{\theta}_1 > 0)$ is one of the three replication probabilities that have been considered by Senn [32] in response to Goodman [33]. The complementary probability $\Pr(\hat{\theta}_2 > 0 \,|\, \hat{\theta}_1 > 0) = 1 - p_{IC}/2$ can be identified as the probability of replicating an effect, $p_{rep}$, advocated by Killeen [13] as an alternative to traditional *p*-values, see Lecoutre & Poitevineau [34] for further discussion and additional references. Of course, $p_{rep}$ is calculated under the assumption that the probability of the null hypothesis ($H_0: \theta = 0$) is zero. Nevertheless, Killeen [35] argues that the use of a predictive procedure such as $p_{rep}$ provides a more positive and productive approach to scientific inference and interpretation than traditional *p*-values.

In practice, we can thus use $p_{IC}$ to assess the probability of replicating an effect, assuming that the null hypothesis is false: $p_{rep} = 1 - p_{IC}/2$. An intrinsically credible result with $p_{IC} \le \alpha$ therefore has $p_{rep} \ge 1 - \alpha/2$. For example, for $\alpha = 5\%$ we have $p_{rep} \ge 97.5\%$. For numerical illustration, recall that the *p*-values for intrinsic credibility in figure 1 are $p_{IC} = 0.021$ (figure 1*a*) and $p_{IC} = 0.09$ (figure 1*b*). The corresponding replication probabilities are thus $p_{rep} = 99.0\%$ and $p_{rep} = 95.5\%$. In the second example, there is thus a $p_{rep} = 4.5\%$ chance that an identically designed replication study will give a negative effect estimate.

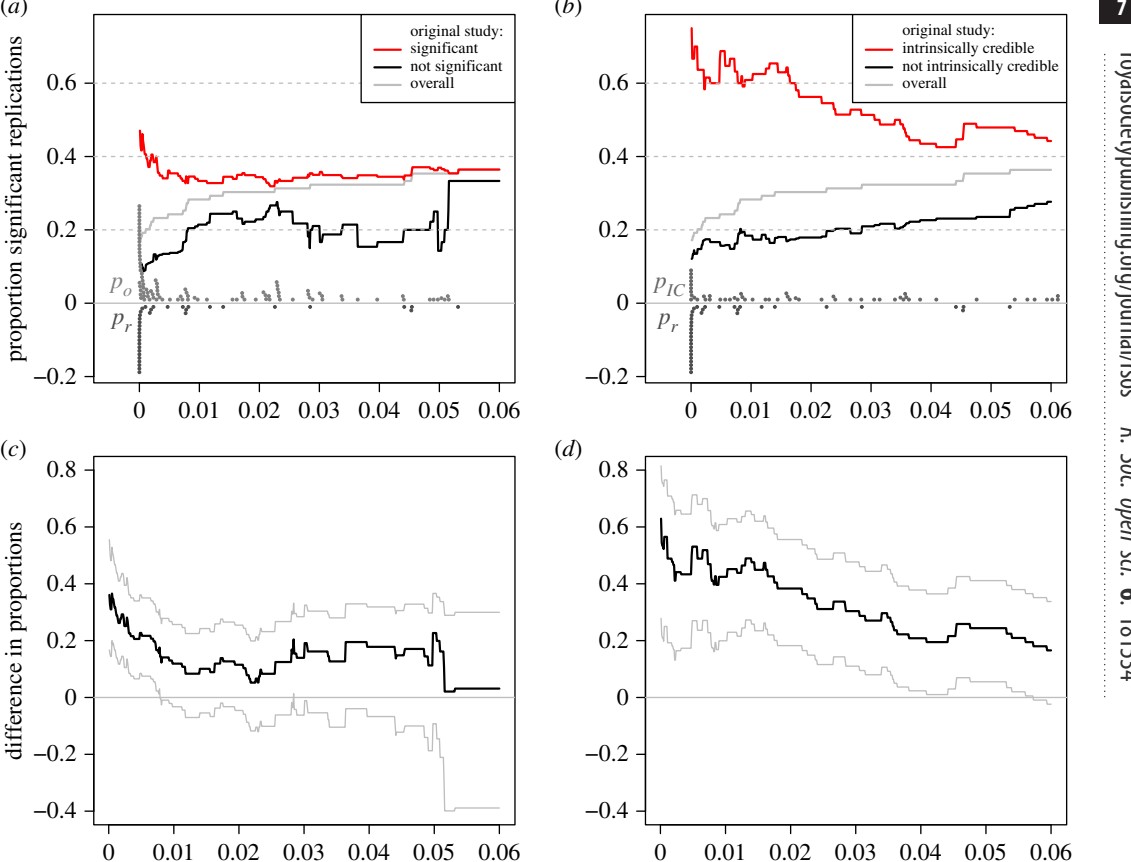

**Figure 4.** Analysis of OSC data. (*a,b*) shows the proportion of significant replication experiments at level $\alpha$, stratified by significant and non-significant original experiments at level $\alpha$ (*a*) and intrinsically credible and not intrinsically credible original experiments at level $\alpha$ (*b*), respectively. The level $\alpha$ varies between 0.0001 and 0.06. The dots at the bottom of plot (*a*) are the corresponding standard $p$-values $p_o$ and $p_r$ from the original (light grey) and replication (dark grey) experiment. The dots at the bottom of plot (*b*) are the $p$-values for intrinsic credibility from the original experiment (light grey) and again the standard $p$-value $p_r$ from the replication experiment (dark grey). (*c,d*) shows the differences between the corresponding proportions with 95% confidence intervals.

## 5. Application

I now reanalyse data from the Open Science Collaboration study on the reproducibility of psychological science [15], where 99 studies report $p$-values for both the original and the replication experiment. First, I have grouped the original studies into significant and non-significant at varying levels of $\alpha$. For each group, I have calculated the proportion of significant replication experiments at the same level $\alpha$. The two proportions are compared in figure 4*a* as a function of $\alpha$. Figure 4*c* shows the difference in the two proportions with 95% confidence intervals. For comparison, figure 4*b* shows the proportions of significant replication experiments (at level $\alpha$) for the groups of intrinsically credible and not intrinsically credible original studies, again for varying levels of $\alpha$. Figure 4*d* displays the difference between these two proportions.

The overall proportion of significant replication experiments increases from 17% to 36% for the range of $\alpha$ values considered (0.0001 to 0.06), see the grey lines in figure 4*a*. The top left plot shows that the significance of the original experiment is a rather poor classifier to predict the success (significance) of the replication experiment for nearly all levels of $\alpha$. Specifically, among the significant original studies, the proportion of significant replications is between 32% and 47%. Based on the confidence intervals for the difference in proportions to the non-significant group shown in the bottom left plot, there is evidence for a true difference only for very small $\alpha$ levels ($\alpha < 0.01$). However, if we use intrinsic credibility as a classifier, the proportion of significant replications is between 43% and 75% and the difference to the other (not intrinsically credible) group is substantially larger for most values of $\alpha$.

Furthermore, the lower bound of the confidence interval for the difference is nearly always positive. These results suggest that intrinsic credibility of the original experiment is better suited to predict the success of a replication experiment than standard significance.

# 6. Discussion

The concept of intrinsic credibility has been proposed to assess claims of new discoveries in the absence of prior evidence. I have shown that a combination of the Analysis of Credibility with the Box [10] check for prior-data conflict directly leads to a more stringent threshold $\alpha_{IC}$ for intrinsic credibility. If one prefers to avoid any thresholding of $p$-values, a new $p$-value for intrinsic credibility, $p_{IC}$, has been proposed. $p_{IC}$ is a quantitative measure of intrinsic credibility with a direct connection to $p_{rep}$, the probability of replicating an effect [13].

The assessment of intrinsic credibility can be thought of as an additional challenge, ensuring that claims of new discoveries are not spurious. Conventionally significant results with $0.05 > p > 0.0056$ lack intrinsic credibility, i.e. they are not in conflict with the sufficiently sceptical prior that would make the effect just non-significant. This matches the classification as 'suggestive' by Benjamin *et al.* [6]. Specifically, $p > 0.0056$ implies $p_{IC} > 0.05$ and thus $p_{rep} < 97.5\%$, emphasizing the need for replication. If $p < 0.0056$, then the result is both significant and intrinsically credible at the 5% level, so $p_{IC} \leq 0.05$ and $p_{rep} \geq 97.5\%$.

The credibility ratio provides a simple and convenient tool to check whether a 'significant' confidence interval at any level $\gamma$ is also intrinsically credible. If the credibility ratio is smaller than 5.8, the result can be considered as intrinsically credible at level $\alpha$. It is noteworthy that the concept of intrinsic credibility does not require changing the original confidence level $\gamma$. Indeed, the check for credibility is done at the same level as the original confidence level. I have used $\gamma = 0.95$ by convention, where it follows that the check for intrinsic credibility is equivalent to the requirement $p < 0.0056$. This implies that in standard statistical reporting there is no need to replace 95% confidence intervals with 99.5% confidence intervals, say. However, for claims of new discoveries, I suggest complementing or replacing the ordinary $p$-value with the proposed $p$-value for intrinsic credibility, $p_{IC}$.

Although derived using a Bayesian approach, the proposed check for intrinsic credibility is based on a standard confidence interval and thus constitutes a Bayes/non-Bayes compromise [36]. Specifically, it does not require the specification of a prior probability of the null hypothesis of no effect. In fact, this prior probability is always zero. This is in contrast to the calibration of $p$-values to lower bounds on the posterior probability of the null, which requires specification of a prior probability. Minimum Bayes factors have also been proposed to calibrate $p$-values, see Held & Ott [37] for a recent review. They have the advantage that they do not require specification of a prior probability of the null hypothesis and provide a direct 'forward-Bayes' assessment of the evidence of $p$-values. However, the underlying rationale is still based on a point null hypothesis with positive prior probability, fundamentally different from the approach proposed here.

The Analysis of Credibility assumes a simple mathematical framework, where likelihood, prior and posterior are all normally distributed. This can be justified because Gaussian approximations are commonly used in the calculation of confidence intervals and statistical hypothesis tests, if the sample size is fairly large (e.g. [29, Section 2.4]). Of course, suitable transformations of the parameter of interest may be needed to achieve normality, for example, confidence intervals for odds ratios and hazard ratios should be transformed to the log scale and Fisher's $z$-transformation should be applied to correlation coefficients. For small studies, however, the normal assumption for the likelihood may be questionable and the assessment of intrinsic credibility would need appropriate refinement, for example, based on the $t$-distribution. It may also be of interest to extend the approach to settings where the sceptical prior is conjugate to a likelihood within the exponential family.

Ethics. No new data have been collected so an approval from an ethics committee or a permission from any other authority was not needed.

Data accessibility. Data analysed in this article are originally from Open Science Collaboration [15] and have been downloaded from https://osf.io/fgjvw/, where the two columns T_pval_USE..O. and T_pval_USE..R. in the file rpp_data.csv represent the $p$-values from the original and replication experiment, respectively. Only those two columns are used.

Competing interests. I declare no competing interests.

Authors' contributions. I am the single author of this manuscript and wrote the manuscript on my own.

Funding. I received no funding for this study.

Acknowledgements. I am grateful to Robert Matthews, Stefanie Muff, Manuela Ott and Kelly Reeve for helpful comments on earlier drafts of this manuscript. I would also like to thank two referees for constructive comments on an earlier version of this paper.

# Appendix

# Proof of equation (2.1) and (3.3)

With $U, L = \hat{\theta} \pm z_{\alpha/2}\sigma$ we have $UL = \hat{\theta}^2 - z_{\alpha/2}^2\sigma^2$ and $U - L = 2z_{\alpha/2}\sigma$. We therefore obtain with equation (2.2) and $\tau = S/z_{\alpha/2}$:

$$\tau^2 = \frac{S^2}{z_{\alpha/2}^2} = \frac{(U-L)^4}{16z_{\alpha/2}^2 UL} = \frac{(2z_{\alpha/2}\sigma)^4}{16z_{\alpha/2}^2 UL} = \frac{z_{\alpha/2}^2\sigma^4}{\hat{\theta}^2 - z_{\alpha/2}^2\sigma^2} = \frac{z_{\alpha/2}^2\sigma^2}{t^2 - z_{\alpha/2}^2} = \frac{\sigma^2}{t^2/z_{\alpha/2}^2 - 1}.$$

To show equation (3.3), we use $\hat{\theta}^2 = UL + z_{\alpha/2}^2\sigma^2$ and $\sigma^2 = (U-L)^2/(4z_{\alpha/2}^2)$ and obtain

$$t_{\text{Box}}^2 = t^2 - z_{\alpha/2}^2 = \frac{\hat{\theta}^2}{\sigma^2} - z_{\alpha/2}^2 = \frac{UL + z_{\alpha/2}^2\sigma^2}{\sigma^2} - z_{\alpha/2}^2 = \frac{UL}{\sigma^2} = z_{\alpha/2}^2\frac{4UL}{(U-L)^2}.$$

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
