## [Reviewer comments · Royal Society Open Science]

Review History

RSOS-181534.R0 (Original submission)

Review form: Reviewer 1 (Miguel de Carvalho)

Is the manuscript scientifically sound in its present form?

Yes

Are the interpretations and conclusions justified by the results?

Yes

Is the language acceptable?

Yes

Is it clear how to access all supporting data?

Yes

Do you have any ethical concerns with this paper?

No

Have you any concerns about statistical analyses in this paper?

Yes

Recommendation?

Accept with minor revision (please list in comments)

Comments to the Author(s)

See attached file (Appendix A).

Review form: Reviewer 2 (Stephen Senn)

Is the manuscript scientifically sound in its present form?

Yes

Are the interpretations and conclusions justified by the results?

Yes

Is the language acceptable?

Yes

Is it clear how to access all supporting data?

Yes

Do you have any ethical concerns with this paper?

No

Have you any concerns about statistical analyses in this paper?

No

Recommendation?

Accept with minor revision (please list in comments)

Comments to the Author(s)

I have only minor comments but raise a couple of matters you may wish to take into account.

1) It is inherent to a pure Bayesian approach that data and prior distribution are exchangeable to the degree defined by the model (1). Now, hardly anybody checks that the second half of a data set is compatible with the first. This thus raises the issue as to why one would check the compatibility of the data and some prior. A pure Bayesian approach would be simply to update one's beliefs and it is not immediately clear to me how you expect the statistician, whether frequentist or Bayesian to use the concept. Are data to be rejected? Are possible prior distributions to be called into account? What philosophy of statistics do either of these correspond to? Of course to raise these comments is perhaps to be a bit purist and practical data analysis is often a complex and "dirty" business. Nevertheless a very brief discussion might be welcome.

2) In my opinion you do not entirely escape a common confusion unnecessarily introduced by Bayesians interpreting P-values in ways they are not meant to be interpreted. In my view the paper by Benjamin et al (2018) is an example of this. P-values have a long history in which they can be reasonably interpreted in one of two ways
a) A one-sided P-value can be interpreted as the Bayesian probability that the true treatment effect is after all negative rather than (say) positive if an uninformative prior distribution is taken to apply. This is the inverse probability

interpretation that Student gave in his paper of 1908 following on from Laplace etc. The term was not in use in 1908 but Student was what would now be described as a Bayesian. An enormous amount of modern applied Bayesian work (rightly or wrongly) uses this sort of analysis. Fisher (although he was not the first to do so) proposed a direct probability interpretation as the probability of a result as extreme or more extreme and also proposed doubling it. Most of the critics of P-values sign up to neither of these interpretations but instead start from the position that what a P-value ought to be is something more along the lines of a probability associated with the Jeffreys hypothesis test. What you propose does not go as far as that. Nevertheless, you are suggesting something more conservative than the conventional P-value. That's fine. However, in my opinion, when you implicitly adopt a classical frequentist calibration without discussion, you go too far. You adopt a classification for the original P-values proposed by Bland (2015) as being appropriate for the modified one. (In fact you somewhat misrepresent Bland who uses 'Evidence' and not 'Moderate evidence' for the range 0.01 to 0.05). However, either Bland's standard is appropriate for P-values, in which case the correct classifications for $P=0.0011$ is 'strong evidence' and the fact that $PIC = 0.021$ does not justify labelling this as merely 'evidence', or Bland's standard is inappropriate in the first place, in which case, why cite it? Is one meant to think 'if only Bland understood the evidential value of P-values he would reclassify his scheme'? However, Bland is a very experienced statistician and his calibration no doubt reflects that experience. In short, in my view, you have committed the mistake of assuming that when a measuring instrument is changed the appropriate numerical thresholds do not (2).

References

1. Senn SJ. Comment on Gelman and Shalizi. *Br J Math Stat Psychol*. 2013;66(1):65-67.
2. Senn S. Double Jeopardy?: Judge Jeffreys Upholds the Law. *Error Statistics Philosophy* 2015; <http://errorstatistics.com/2015/05/09/stephen-senn-double-jeopardy-judge-jeffreys-upholds-the-law-guest-post/>. Accessed 13 February, 2016.

Decision letter (RSOS-181534.R0)

25-Jan-2019

Dear Dr Held

On behalf of the Editors, I am pleased to inform you that your Manuscript RSOS-181534 entitled "The Assessment of Intrinsic Credibility and a New Argument for $p < 0.005$ " has been accepted for publication in Royal Society Open Science subject to minor revision in accordance with the referee suggestions. Please find the referees' comments at the end of this email.

The reviewers and handling editors have recommended publication, but also suggest some minor revisions to your manuscript. Therefore, I invite you to respond to the comments and revise your manuscript.

- Ethics statement

- Data accessibility

It is a condition of publication that all supporting data are made available either as supplementary information or preferably in a suitable permanent repository. The data

accessibility section should state where the article's supporting data can be accessed. This section should also include details, where possible of where to access other relevant research materials such as statistical tools, protocols, software etc can be accessed. If the data has been deposited in an external repository this section should list the database, accession number and link to the DOI for all data from the article that has been made publicly available. Data sets that have been deposited in an external repository and have a DOI should also be appropriately cited in the manuscript and included in the reference list.

If you wish to submit your supporting data or code to Dryad (<http://datadryad.org/>), or modify your current submission to dryad, please use the following link:
<http://datadryad.org/submit?journalID=RSOS&manu=RSOS-181534>

- **Competing interests**

- **Authors' contributions**

- **Acknowledgements**

- **Funding statement**

Because the schedule for publication is very tight, it is a condition of publication that you submit the revised version of your manuscript before 03-Feb-2019. Please note that the revision deadline will expire at 00.00am on this date. If you do not think you will be able to meet this date please let me know immediately.

To revise your manuscript, log into <https://mc.manuscriptcentral.com/rsos> and enter your Author Centre, where you will find your manuscript title listed under "Manuscripts with

Decisions". Under "Actions," click on "Create a Revision." You will be unable to make your revisions on the originally submitted version of the manuscript. Instead, revise your manuscript and upload a new version through your Author Centre.

Once again, thank you for submitting your manuscript to Royal Society Open Science and I look

forward to receiving your revision. If you have any questions at all, please do not hesitate to get in touch.

Kind regards,
 Andrew Dunn
 Senior Publishing Editor
 Royal Society Open Science
 openscience@royalsociety.org

on behalf of Prof Mark Chaplain (Subject Editor)
 openscience@royalsociety.org

Reviewer comments to Author:

Reviewer: 1

Comments to the Author(s)

See attached file

Reviewer: 2

Comments to the Author(s)

I have only minor comments but raise a couple of matters you may wish to take into account.
 1) It is inherent to a pure Bayesian approach that data and prior distribution are exchangeable to the degree defined by the model (1). Now, hardly anybody checks that the second half of a data set is compatible with the first. This thus raises the issue as to why one would check the compatibility of the data and some prior. A pure Bayesian approach would be simply to update one's beliefs and it is not immediately clear to me how you expect the statistician, whether frequentist or Bayesian to use the concept. Are data to be rejected? Are possible prior distributions to be called into account? What philosophy of statistics do either of these correspond to? Of course to raise these comments is perhaps to be a bit purist and practical data analysis is often a complex and "dirty" business. Nevertheless a very brief discussion might be welcome.

2) In my opinion you do not entirely escape a common confusion unnecessarily introduced by Bayesians interpreting P-values in ways they are not meant to be interpreted. In my view the paper by Benjamin et al (2018) is an example of this. P-values have a long history in which they can be reasonably interpreted in one of two ways

a) A one-sided P-value can be interpreted as the Bayesian probability that the true treatment effect is after all negative rather than (say) positive if an uninformative prior distribution is taken to apply. This is the inverse probability interpretation that Student gave in his paper of 1908 following on from Laplace etc. The term was not in use in 1908 but Student was what would now be described as a Bayesian. An enormous amount of modern applied Bayesian work (rightly or wrongly) uses this sort of analysis.

b) Fisher (although he was not the first to do so) proposed a direct probability interpretation as the probability of a result as extreme or more extreme and also proposed doubling it.

Most of the critics of P-values sign up to neither of these interpretations but instead start from the position that what a P-value *ought* to be is something more along the lines of a probability associated with the Jeffreys hypothesis test. What you propose does not go as far as that. Nevertheless, you are suggesting something more conservative than the conventional P-value. That's fine. However, in my opinion, when you implicitly adopt a classical frequentist calibration without discussion, you go too far. You adopt a classification for the original P-values proposed by Bland (2015) as being appropriate for the modified one. (In fact you somewhat

misrepresent Bland who uses 'Evidence' and not 'Moderate evidence' for the range 0.01 to 0.05). However, either Bland's standard is appropriate for P-values, in which case the correct classifications for $P=0.0011$ is 'strong evidence' and the fact that $PIC = 0.021$ does not justify labelling this as merely 'evidence', or Bland's standard is inappropriate in the first place, in which case, why cite it? Is one meant to think 'if only Bland understood the evidential value of P-values he would reclassify his scheme'? However, Bland is a very experienced statistician and his calibration no doubt reflects that experience. In short, in my view, you have committed the mistake of assuming that when a measuring instrument is changed the appropriate numerical thresholds do not (2). 

References

1. Senn SJ. Comment on Gelman and Shalizi. *Br J Math Stat Psychol*. 2013;66(1):65-67.

2. Senn S. Double Jeopardy?: Judge Jeffreys Upholds the Law. *Error Statistics Philosophy* 2015; <http://errorstatistics.com/2015/05/09/stephen-senn-double-jeopardy-judge-jeffreys-upholds-the-law-guest-post/>. Accessed 13 February, 2016.

Author's Response to Decision Letter for (RSOS-181534.R0)

See Appendix B.

Decision letter (RSOS-181534.R1)

13-Feb-2019

Dear Dr Held,

I am pleased to inform you that your manuscript entitled "The Assessment of Intrinsic Credibility and a New Argument for $p<0.005$ " is now accepted for publication in Royal Society Open Science.

on behalf of Prof Mark Chaplain (Subject Editor)
openscience@royalsociety.org

Appendix A

The Assessment of Intrinsic Credibility and a New Argument for $p < 0.05$.

The paper presents a new threshold for intrinsic credibility, along with a corresponding p -value, and offers a new case in favor of the recently advocated rule $p < 0.05$. The bulk of the main contribution is documented in §§3–4, and an application is offered on §5. There is a lot to like about this manuscript, it is interesting, it goes straight to the point, and it fairly acknowledges shortcomings with the analysis. I have some recommendations on some points that could streamline and enrich the document.

- *Background and Goals*: I found §2 to be clear on describing background, and §§3–4 to be also relatively clear at presenting the main tools; and yet the abstract and Introduction are much more convolved, and are not sufficiently clear about what one will find in the manuscript (nor on the main contributions). I enjoy the motivation in the Introduction—in terms of interest on the problem—but think more concrete details could be anticipated on what are the main contributions on the manuscript, as well as on why these contributions are important.
- *On Geometrical Principles of Compatibility*: The marginal likelihood resulting from a prior-predictive sceptical-prior (π_S) can be regarded as an inner product between π_S and the likelihood (ℓ), and thus it can be interpreted as an angle on the Hilbert space $(L_2(\Theta), \langle \cdot, \cdot \rangle)$ (de Carvalho et al., 2018, in press). I suggest bringing on board some intuition and brief remarks on the geometrical principles of compatibility—understood here as formally defined in Definition 2 and its variants on §3 on the latter paper—so to enrich the discussion on §3. In addition, I wonder about:
- *Compatibility Between Sceptical Prior and the Distribution of $\hat{\theta}$* : It seems natural asking whether compatibility between the posterior (p) and the distribution of $\hat{\theta}$ is of any relevance, or whether one is only concerned with compatibility between π_S and data?
- *Robustness of Claims to Misspecification*: Keeping in mind that the paper makes the bold statement on a new argument for $p < 0.05$, one wonders: “How robust is Fig. 3 to model misspecification?” Indeed, how robust are the claims on the manuscript to model model misspecification? In addition:
- *Transformations / Reparametrizations*: On p. 17 a comment is made on the possible need for working with a transformation $g(\theta)$. Some further remarks would be welcome on the consequences that these transformations could have, as well as on the possibility of having to work with reparametrizations of a model.

Minor comments:

- p. 5: It should be made clear from the onset that it is being assumed that $\hat{\theta} \sim N(\theta, \sigma^2)$ and $\theta \sim N(0, \tau^2)$, instead of one having to collect these separate pieces of information throughout §2.
- p. 6: It would be worth connecting the text and Fig. 1 in terms of conclusions on intrinsic credibility (or lack of it); currently, after the notion of intrinsic credibility is defined on p. 6 no reference is made to Fig. 1 (certainly the relevant information appears on the titles of Fig. 1; still I think it would be worth briefly mentioning this in the text after introducing the notion).
- p. 8: I agree with comment on coherency, and I remark that Hartigan’s maximum likelihood prior (Hartigan et al., 1998) is another example of a prior that can be regarded as a data-based prior in a compatibility-based context, and so is the max-compatible prior (de Carvalho et al., 2018, in press, §2.4).
- p. 9: On the definition of credibility ratio, perhaps a remark should be added reminding L will not be zero.
- p. 17: The comment acknowledging the simple mathematical framework is appropriate. Yet I suspect more could be said on the setting where the likelihood is on the exponential family; certainly not part of this paper but perhaps worth remarking.

References

de Carvalho, M., Page, G. L., and Barney, J., B. (2018, in press), “On the geometry of Bayesian Inference,” *Bayesian Analysis*.
Hartigan, J. et al. (1998), “The maximum likelihood prior,” *The Annals of Statistics*, 26, 2083–2103.

Appendix B

The Assessment of Intrinsic Credibility and a New Argument for $p < 0.005$

I am very grateful for the constructive comments made by the referees. I have tried to integrate all suggested changes and have also made some minor additional changes and additions to the manuscript in order to improve clarity. All changes are marked in red in the submitted document.

Response to referee 1

Minor comments:

1. *Background and Goals:* I found §2 to be clear on describing background, and §§3-4 to be also relatively clear at presenting the main tools; and yet the abstract and Introduction are much more convolved, and are not sufficiently clear about what one will find in the manuscript (nor on the main contributions). I enjoy the motivation in the Introduction—in terms of interest on the problem—but think more concrete details could be anticipated on what are the main contributions on the manuscript, as well as on why these contributions are important.

I have added a few sentences on the main contributions of the manuscript and their relevance in Section 1.

2. *On Geometrical Principles of Compatibility:* The marginal likelihood resulting from a prior-predictive sceptical-prior (π_S) can be regarded as an inner product between π_S and the likelihood (l), and thus it can be interpreted as an angle on the Hilbert space (de Carvalho et al., 2018, in press). I suggest bringing on board some intuition and brief remarks on the geometrical principles of compatibility—understood here as formally defined in Definition 2 and its variants on §3 on the latter paper—so to enrich the discussion on §3.

Thank you for this comment. I have added a reference to the work by de Carvalho et al (2018) as an alternative approach to investigate the compatibility of prior and data at the beginning of Section 3. I found the paper by de Carvalho et al (2018) very interesting but felt that a deeper discussion of it would perhaps distract the reader from the main contribution of my manuscript.

In addition, I wonder about:

3. *Compatibility Between Sceptical Prior and the Distribution of $\hat{\theta}$:* It seems natural asking whether compatibility between the posterior (p) and the distribution of $\hat{\theta}$ is of any relevance, or whether one is only concerned with compatibility between π_S and data?

This is an interesting suggestion, but I am not convinced that this is of

major interest. The sceptical prior is centered at zero and the lower or upper credible limit of the posterior is also zero, I therefore do not expect major conflict between the two for any type of data.

4. *Robustness of Claims to Misspecification:* Keeping in mind that the paper makes the bold statement on a new argument for $p < 0.005$, one wonders: “How robust is Fig. 3 to model misspecification?” Indeed, how robust are the claims on the manuscript to model model misspecification?

There was some brief discussion at the end of the discussion section on deviations from the normal assumption which has now been extended.

In addition:

5. *Transformations / Reparametrizations:* On p. 17 a comment is made on the possible need for working with a transformation $g(\theta)$. Some further remarks would be welcome on the consequences that these transformations could have, as well as on the possibility of having to work with reparametrizations of a model.

Some further remarks have been added.

Minor comments:

1. p. 5: It should be made clear from the onset that it is being assumed that $\hat{\theta} \sim N(\theta, \sigma^2)$ and $\theta \sim N(0, \tau^2)$, instead of one having to collect these separate pieces of information throughout §2.

Thanks, done as suggested.

2. p. 6: It would be worth connecting the text and Fig. 1 in terms of conclusions on intrinsic credibility (or lack of it); currently, after the notion of intrinsic credibility is defined on p. 6 no reference is made to Fig. 1 (certainly the relevant information appears on the titles of Fig. 1; still I think it would be worth briefly mentioning this in the text after introducing the notion).

Thanks, done as suggested.

3. p. 8: I agree with comment on coherency, and I remark that Hartigan's maximum likelihood prior (Hartigan et al., 1998) is another example of a prior that can be regarded as a data-based prior in a compatibility-based context, and so is the max-compatible prior (de Carvalho et al., 2018, in press, §2.4).

Thanks, these (and further) references have been added as examples of data-based priors.

4. p. 9: On the definition of credibility ratio, perhaps a remark should be added reminding L will not be zero.

Thanks, added.

5. p. 17: The comment acknowledging the simple mathematical framework is appropriate. Yet I suspect more could be said on the setting where the likelihood is on the exponential family; certainly not part of this paper but perhaps worth remarking.

Thank you, a corresponding comment has been added at the end.

References

de Carvalho, M., Page, G. L., and Barney, J., B. (2018, in press), “On the geometry of Bayesian Inference,” Bayesian Analysis.

Hartigan, J. et al. (1998), “The maximum likelihood prior,” The Annals of Statistics, 26, 20832103.

Response to referee 2

I have only minor comments but raise a couple of matters you may wish to take into account.

1. It is inherent to a pure Bayesian approach that data and prior distribution are exchangeable to the degree defined by the model (1). Now, hardly anybody checks that the second half of a data set is compatible with the first. This thus raises the issue as to why one would check the compatibility of the data and some prior. A pure Bayesian approach would be simply to update one’s beliefs and it is not immediately clear to me how you expect the statistician, whether frequentist or Bayesian to use the concept. Are data to be rejected? Are possible prior distributions to be called into account? What philosophy of statistics do either of these correspond to? Of course to raise these comments is perhaps to be a bit purist and practical data analysis is often a complex and ”dirty” business. Nevertheless a very brief discussion might be welcome.

Thank you for your comment. Box (1980) makes the distinction between model estimation and model criticism and provides a principled approach for model criticism based on the prior predictive distribution. In fact, Fisher’s significance test turns out to be a special case of the model criticism approach (Section 2.1 in Box, 1980). I agree with the reviewer that this is not part of the pure Bayesian approach to inference, but I feel (as Box) that this is in fact a major weakness of it. The discussion of Box (1980) provides further discussion of this issue and I have added a corresponding remark in the paper.

2. In my opinion you do not entirely escape a common confusion unnecessarily introduced by Bayesians interpreting P-values in ways they are not meant to be interpreted. In my view the paper by Benjamin et al (2018) is an example of this. P-values have a long history in which they can be reasonably interpreted in one of two ways a) A one-sided P-value can be interpreted as the Bayesian probability that the true treatment effect is after all negative rather than (say) positive if an uninformative prior distribution is taken to apply. This is the inverse probability interpretation that Student gave in his paper of 1908 following on from Laplace etc. The term was not in use in 1908 but Student was what would now be described as a Bayesian. An enormous amount of modern applied Bayesian work (rightly or wrongly) uses this sort of analysis. b) Fisher (although he was not the first to do so) proposed a direct probability interpretation as the probability of a result as extreme or more extreme and also proposed doubling it. Most of the critics of P-values sign up to neither of these interpretations but instead start from the position that what a P-value *ought* to be is something more along the lines of a probability associated with the Jeffreys hypothesis test. What you propose does not go as far as that. Nevertheless, you are suggesting something more conservative than the conventional P-value. That's fine. However, in my opinion, when you implicitly adopt a classical frequentist calibration without discussion, you go too far. You adopt a classification for the original P-values proposed by Bland (2015) as being appropriate for the modified one. (In fact you somewhat misrepresent Bland who uses 'Evidence' and not 'Moderate evidence' for the range 0.01 to 0.05). However, either Bland's standard is appropriate for P-values, in which case the correct classifications for $P=0.0011$ is 'strong evidence' and the fact that $PIC = 0.021$ does not justify labelling this as merely 'evidence', or Bland's standard is inappropriate in the first place, in which case, why cite it? Is one meant to think 'if only Bland understood the evidential value of P-values he would reclassify his scheme'? However, Bland is a very experienced statistician and his calibration no doubt reflects that experience. In short, in my view, you have committed the mistake of assuming that when a measuring instrument is changed the appropriate numerical thresholds do not (2).

I agree with the reviewer and have deleted the interpretation of the proposed p-values for intrinsic credibility based on Bland's categories.

References

1. Senn SJ. Comment on Gelman and Shalizi. Br J Math Stat Psychol. 2013;66(1):65-67.
2. Senn S. Double Jeopardy?: Judge Jeffreys Upholds the Law. Error Statistics Philosophy 2015; <http://errorstatistics.com/2015/05/09/stephen-senn-double-jeopardy-judge-jeffreys-upholds-the-law-guest-post/>. Accessed 13 February, 2016.